# Efficacy of laparotomy sponges to reduce bacterial contamination using an in vitro gastrointestinal surgery model

**Alla Bezhentseva**[ID]*, **Lindsay L. St. Germaine, Daniel E. Hoffmann**

Surgery Department, Veterinary Specialists & Emergency Services, Rochester, NY, United States of America

* abezh27@gmail.com

**Data Availability Statement:** All images and result files are available from the OSF database. https://osf.io/fhdu7/?view_only=98e3005fe4d14319b508ac34b538de18.

## Abstract

Use of laparotomy sponges to protect abdominal viscera during gastrointestinal surgery is described in nonspecific terms by various sources, but no definitive guidelines have been established in veterinary literature. The objective of this study was to compare the in vitro efficacy of various layer-densities of laparotomy sponges at reducing bacterial contamination from multiple contaminant volumes during multiple exposure times. A standardized *Escherichia coli* inoculum water solution was applied over sterile laparotomy sponges overlying blood agar plates. Four laparotomy sponge layer-densities, 4 volumes of *E. coli* inoculum water solution, and 4 exposure times were evaluated. All blood agar plates were incubated for 48 hours followed by surface area measurements of colonization of each blood agar plate at 24 and 48 hours. The procedure was repeated thrice. Bacterial colonization occurred on 100% (192/192) of inoculated blood agar plates. There was a statistically significant decrease in colonized area with increasing layer-density of laparotomy sponges (P<0.0001). Comparison between the layer-density of sponges were statistically significant in resulting infected area (P<0.01), except comparison between 6- and 8-layers (P = 0.9490). Colonized area was not significantly altered by time of exposure. Results suggested that increasing the layer-density of laparotomy sponges has significant effect on reducing strikethrough bacterial colonization in an in vitro model. The results of this study can be used when performing gastrointestinal surgery to help guide laparotomy sponge use to reduce peritoneal bacterial contamination.

## Introduction

Gastrointestinal (GI) surgery is performed by small animal practitioners in numerous settings including specialty practices, primary care practices, and shelter medicine, etc. The indications for GI surgery include recovery of foreign material causing obstruction, intussusception, excisional or incisional biopsies of neoplastic lesions, and obtaining full-thickness intestinal biopsies [1, 2]. In addition to protecting surrounding structures from contamination with contents and microflora, aseptic technique is necessary when making an incision into the hollow viscus (i.e., GI organs) [2]. Methods include the use of moistened laparotomy sponges (further

**Funding:** Laparotomy sponges were provided by Everost Veterinary Orthopedics through their relationship with STERIS Animal Health. The funders had no role in study design, data collection and analysis, decision to publish, or preparation of the manuscript. Blood agar plates were provided by Monroe Veterinary Associates through their relationship with Veterinary Laboratory of Rochester. The funders had no role in study design, data collection and analysis, decision to publish, or preparation of the manuscript.

**Competing interests:** The authors have declared that no competing interests exist.

referred to by the term sponges) to cover the abdominal wall incision, using sponges to isolate the affected segment from the remainder of the abdomen, and by exteriorizing freely movable segments out of and away from the incision [2].

Overflow of GI contents during gastrotomy, enterotomy, or intestinal resection & anastomosis (IRA) may be contained within sponges if the segment was successfully exteriorized, or the volume is small. With greater volumes, there is an increased risk of contact with the dermis, subcutaneous tissues, body wall, or abdominal viscera, resulting in contamination and potential morbidity. Contamination may increase the risk of infection, cause significant inflammation, result in increased cost, prolonged hospitalization, and increased morbidity and mortality. A recent publication [3] looking at surgical site infections following GI surgery found that 7% of patients developed a surgical site infection, with the most common bacterial isolate of *Escherichia coli*. Within the group of patients, the majority had bacterial isolates that were not susceptible to the antibiotics used intraoperatively (cefazolin, cefoxitin) [3].

Canine GI microflora has been studied using biopsy and fecal samples of both healthy and diseased patients. Each segment of the healthy GI tract varies in the colony-forming units (CFU) per gram of sampled tissue, increasing from $10^4$–$10^5$ CFU/g to $10^9$–$10^{11}$ CFU/g from stomach to colon, respectively [4]. Based on sequencing studies of biopsy samples from healthy patients, Proteobacteria occupy 21.2% to 26.6% of duodenal tissue, 46.7% of jejunal tissue, and only 1.4% within the colon [4]. The healthy microbiome can be altered with diet, disease states, the use of antibiotics, and/or pre/probiotics [4–6]. Common consequences of these alterations in man and laboratory animals are increased ratios of pathogenic microbes, such as *E. coli*, *Salmonella*, *Proteus*, *Klebsiella*, and *Shigella* [5]. In addition, commensal microbes, such as the Proteobacteria *E. coli*, and others have been linked to causes of some inflammatory GI diseases [7]. In patients presenting for GI surgery due to a disease-state, dysbiosis of commensal microbial populations may be occurring. If dysbiosis is severe enough to allow for pathogenic or multi-drug resistant strains to predominate, the surgeon should recognize the potential for bacterial contamination and its sequelae. Development of bacterial peritonitis secondary to dehiscence of small intestinal surgery (enterotomy, IRA) is reported to occur in 7% to 16% historically [2]. Recently, dehiscence rates associated with enterotomy versus IRA for retrieval of intestinal foreign bodies were reported to be 3.8% and 18.2%, respectively, with an overall rate of 6.6% [8]. The resulting septic peritonitis necessitates source control via surgical intervention, peritoneal lavage, and antimicrobial therapies directed at suspected bacterial isolates [2].

To the authors' knowledge, there are no peer-reviewed veterinary publications that establish guidelines for use of laparotomy sponges for GI surgical procedures. Veterinary textbooks are limited to recommending that surgical sites are "walled off with" or "packed off with" moistened sponges followed by local or peritoneal lavage [2, 9–11]. As such, research is needed to evaluate the efficacy of laparotomy sponges used for GI surgeries. The purpose of this study is to compare the in vitro efficacy of various layer-densities of sponges at reducing contamination with exposure to multiple contaminant volumes during multiple exposure times. Our null hypothesis is that the degree of bacterial contamination will not be dependent upon layer-density of sponges, volume of contaminant, time of exposure to contaminant, or any combination thereof.

## Materials and methods

### Materials preparation

A non-pathogenic quality-control strain of *E. coli*, contained within a manufactured bacteriological loop (Culti-Loops *Escherichia coli* ATCC, Thermo Scientific, Lenexa, Kansas, USA),

was streaked onto a blood agar plate (Blood agar (TSA w/ sheep blood) plate, Remel, Lenexa, Kansas, USA) by a microbiology technician and incubated at 37˚C for 24 hours. If bacterial colonies were visible, the plate was then refrigerated at approximately 4˚C until testing commenced. If no bacterial colonies were visible, the plate was discarded, and a new bacteriological loop was streaked onto a new blood agar plate and incubation was repeated. This original cultivation was then used as follows. Twenty-four hours prior to each testing day, a pre-flamed inoculating loop was used to obtain an *E. coli* colony from the original cultivation and was streaked onto a new blood agar plate. After 24 hours of incubation, *E. coli* inoculated water (further referred to as *E. coli* water) was created to simulate gastric or intestinal contents as follows: single-use wooden applicators were used to obtain and place colonies within 3ml inoculum water vials (MicroScan Inoculum Water, Beckman Coulter Inc., Brea, California, USA) without disturbing the underlying plate medium. A table-top turbidity meter (MicroScan Tubidity Meter, Siemens Healthcare Diagnostics Inc., West Sacramento, California, USA), calibrated daily to 0.5 McFarland Standard turbidity ($1.5X10^8$ bacterial suspension/ml equivalent) [12], was used to compare each inoculated vial to a new inoculum water vial. This McFarland Standard was chosen due to its approximation to previously described microbial counts within the small and large intestines [4–7]. Colonies were added to each vial until 0.5 McFarland Standard was achieved for each inoculated vial. The process was repeated until the pre-calculated total volume is obtained (26 vials or 78ml). Laparotomy sponges (30.5cm x 30.5cm) (Dukal Corporation, Ronkonkoma, New York, USA), tongue depressors, and patient drapes were prepared using standard sterilization methods. A 4-by-4 grid was created to assign numbers to each plate for standardization across all trials (Fig 1). All new plates were numbered prior to testing.

**Fig 1. Grid representing plate placement for experimental procedure.** Plate numbers (#1–16) were standardization across all trials. The left column indicates the volume of *E. coli* water applied within each row of plates. The top row indicates the time of exposure within each column of plates. The number assigned to each control plates using the random number generator followed this grid to dictate the variables to be tested in the control setting.

## Experimental procedure

Using aseptic technique in an operating room, the investigator prepared a plating table and an instrument table each covered with a sterile patient drape (STERIS Animal Health, Birmingham, Alabama, USA). All required sterile tongue depressors, syringes, a large stainless-steel bowl, and sponges were obtained and placed on the instrument table using aseptic technique (S1 Fig). An assistant poured 1L of sterile 0.9% sodium chloride irrigation solution into the large sterile stainless-steel bowl and sponges were soaked, squeezed out manually, and each folded into the layer-density being evaluated per trial. For 2 layers, 1 sponge was folded in half once. For 4 layers, 1 sponge was folded in half twice. For 6 layers, 1 sponge was folded in half twice and a second sponge was folded in half once and these were stacked. For 8 layers, 2 sponges were each folded in half twice and these were stacked. The total volume (78ml) of *E. coli* water was emptied by an assistant into a small sterile stainless-steel bowl opened aseptically onto a mayo stand. The numbered new blood agar plates were laid out by the assistant into the 4-by-4 grid onto the plating table.

At the start of each trial, the assistant opened the lid of the blood agar plates in the 30s group (Fig 1) and laid the lid with the inside facing down to the left of each plate. This method was used to prevent any air-borne contamination of the plate once the lid was replaced. Without breaking sterility, the folded sponges were placed over the open blood agar plate so that one sponge surface is in contact with the blood agar. When testing 2 layers and 6 layers, the excess of the sponge folded in half once was allowed to drape over the edge of the blood agar plate onto the patient drape. Sterile tongue-depressors were used to gently press sponges down onto the plate to ensure gentle contact of one surface of the folded sponge(s) with the blood agar, and then discarded. The *E. coli* water was gently agitated within the bowl prior to each draw like the preparation method. The volume of *E. coli* water being evaluated was drawn up and applied to the center of the sponge over the plate. A stopwatch was started when *E. coli* water contacted the sponge on plate 1. At the end of each exposure time group (30s, 120s, 300s, or 600s), sponges were removed using sterile tongue depressors in the same order as they were applied, and both were discarded. The lids were replaced using sterile tongue depressors which were then discarded. New syringes were obtained for each exposure time group. This procedure was repeated until all plates on the grid were inoculated, including the control plate. The stopwatch was stopped at the completion of each trial. Each trial was repeated until each sponge layer-density was evaluated three times.

To ensure no cross-contamination between trials, new sterile surgical gloves, patient drapes, sponges, tongue depressors, and syringes were used for each trial. At the completion of each trial, all plates (17 total) were incubated at 37°C for 48 hours. All plates were evaluated and photographed (Canon PowerShot SD1100 IS, Ota City, Tokyo, Japan) at 24 hours of incubation and 48 hours of incubation (S2 Fig).

## Control group

A random number generator (Google, Mountain View, California, USA) was used prior to the start of each trial to establish a control plate corresponding to one of the 16 factor combinations. During setup, the amount of sterile 0.9% sodium chloride irrigation solution required for the control plate inoculation was drawn up in a new syringe and laid aside in the on the instrument table until needed. No *E. coli* water was used for any control plates. The remainder of the procedure was performed as described. If growth on a control plate was identified, the associated trial data was discarded, and the trial was repeated.

## Data acquisition

Photos of all incubated plates were uploaded to a photo-editing software (Adobe Photoshop Desktop, version 22.0, Adobe Inc., San Jose, California, USA). Surface areas of blood agar medium (Plate Area) and bacterial colonization (Colony Area) were measured in pixels. The percentage of bacterial colony occupation of the blood agar plate was then calculated using the formula $\left[\frac{Colony\ Area}{Plate\ Area}\right] x\ 100$. This method was chosen to standardize results across all images. This calculation was performed for each of the 192 plates analyzed at both 24 and 48 hours of incubation (S3 Fig).

## Statistical analysis

The Response Variable (Y) was area infected. There were three factors (X) that could affect this Response Variable: Layers, Time, and Volume. The data were analyzed by means of a Three-Factor Analysis of Variance. The residuals were normally distributed as assessed by means of a histogram and a normal probability plot. Homogeneity of variances was accepted as assessed by means of a plot of the residuals vs. the group means. Post hoc comparisons were by means of Bonferroni *t* test for multiple comparisons. P < 0.05 was considered significant. Data were reported as Mean +/- SEM.

## Results

A total of 240 plates were inoculated with *E. coli* water. Forty-eight inoculated plates were discarded due to growth on the control plate for 3 separate trials and these 3 trials were repeated. A total of 192 plates were included in statistical analysis. Macroscopic bacterial colonization (i.e., area infected) was identified in 100% (192/192 plates) at 24 hours and 48 hours of incubation. The remainder of the results are reported on analysis of data obtained after 48 hours of incubation. Increasing time of exposure of each plate to a contaminated sponge did not significantly increase infected area (P = 0.547). Comparisons of time with layers (P = 0.492), time with volume (P = 0.611), and the three factors together (P = 0.965), did not identify any significant effects of time on area infected.

Infected area was significantly affected by the volume of *E. coli* water (P<0.0001) and the layers of sponges used (P<0.0001). A visual representation of the least square means of these factors and their variable interactions are represented in Fig 2. When comparing the number of layers as the independent factor (2 vs. 4 layers, 2 vs. 6 layers, etc.), all were statistically significant (P<0.01), except comparison between six and eight layers (P = 0.9490). All comparisons of volumes of *E. coli* water as the independent factor (1ml vs. 3ml, 1ml vs. 5ml, etc.) were statistically significant (P<0.0005). The interaction of layers and volume as a group was not statistically significant (P = 0.056). However, individual analysis of the interaction between layers and volume identified 48 of the 120 total comparisons to be important and 25 comparisons being statistically significant. Infected area decreases as the layer of sponges increase for all volumes in a parallel linear fashion. The importance of this interaction is the decrease in infected area is greater as volume increases (10ml vs. 1ml). The statistically significant data is presented in Table 1.

## Discussion

The primary purpose of the present study was to evaluate the efficacy of laparotomy sponges in reducing bacterial contamination in an in vitro model. Due to the prevalence of *E. coli* in the canine gut [4–7], a non-pathogenic strain of *E. coli*, used for quality control measures, was chosen for experimentation. We partially rejected our null hypothesis that bacterial

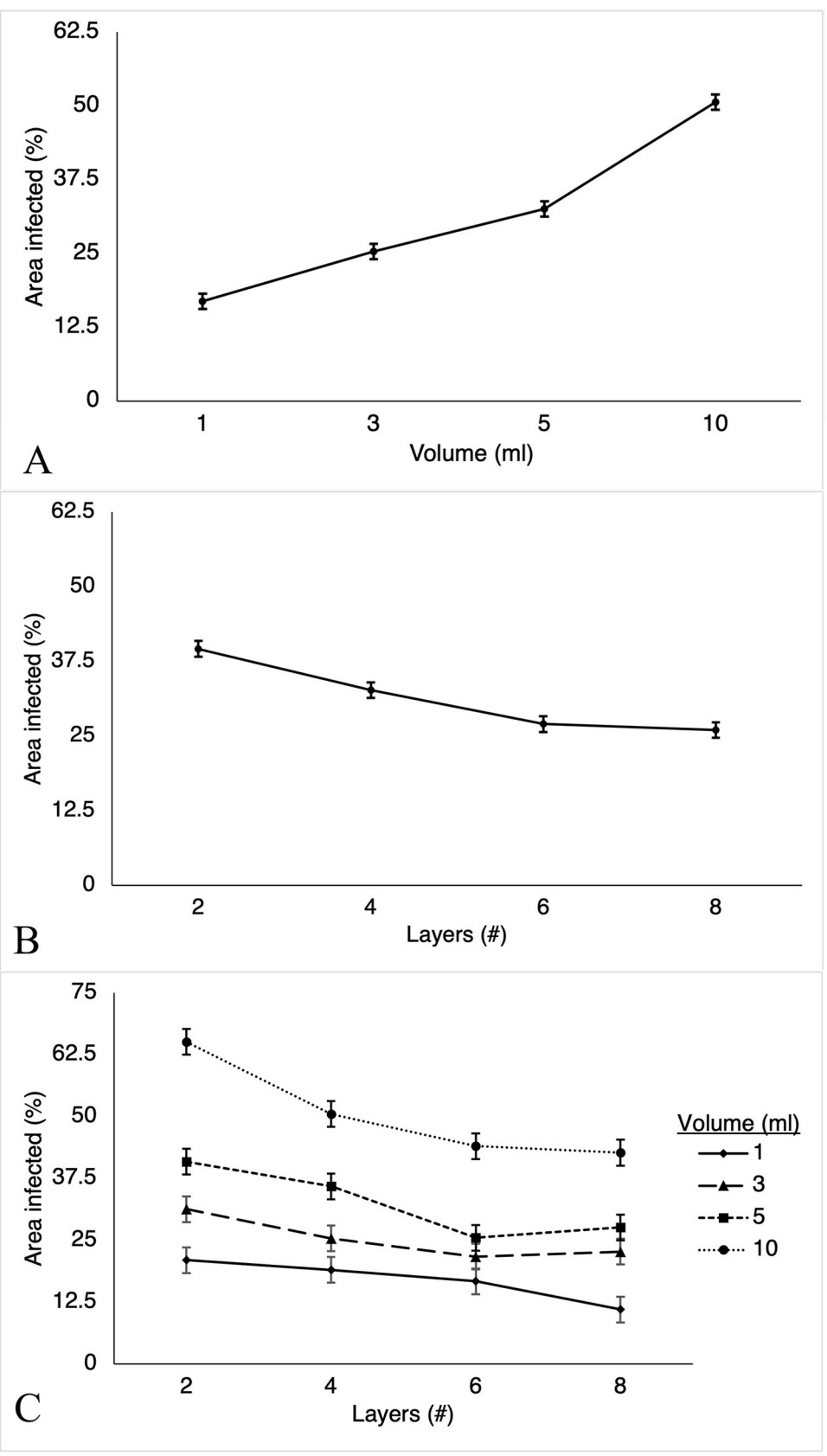

**Fig 2. Least Square Means (LSM) plots for factors impacting infected area on plates.** Increased y-axis values indicate greater infected area on inoculated plates after 48 hours of incubation, reported as a percentage of blood agar medium occupied. X-axis values indicate the variables of statistically significant test factors being reported. Each LSM data point is bound by standard error. LSM for experimental data is identical to the true Means. A–LSM plot of infected area by Volume of *E. coli* water. B–LSM plot of infected area by Layers of laparotomy sponges. C–LSM plot of infected area by the interaction of Layers and Volume.

contamination is not dependent upon layer-density of laparotomy sponges or volume of contaminant. We accepted that time of exposure to contaminant does not impact bacterial colonization. In this in vitro study, growth occurred on 192/192 (100%) of inoculated plates to varying degrees at all investigated sponge layer-densities, occurring with even the minimum *E. coli* water volume and maximum sponge layer-density combination (1ml with 8 layers). An expected finding of this study was that increasing the volume of the contaminant caused increasing areas of bacterial colonization. In addition, increasing the number of layers of

**Table 1. Statistically significant comparisons in post hoc analysis comparing the effect of layers and volume on area infected.**

| Factors | | | | LSM difference | SE | t | P-value[a] |
|---|---|---|---|---|---|---|---|
| Layer | Volume | Layer | Volume | | | | |
| 2 | 1 | 8 | 1 | 0.1000 | 0.0365 | 2.74 | 0.0420 |
| 2 | 5 | 6 | 5 | 0.1544 | 0.0365 | 4.23 | 0.0000 |
| 2 | 5 | 8 | 5 | 0.1336 | 0.0365 | 3.66 | 0.0024 |
| 4 | 5 | 6 | 5 | 0.1036 | 0.0365 | 2.84 | 0.0312 |
| 2 | 10 | 4 | 10 | 0.1464 | 0.0365 | 4.01 | 0.0006 |
| 2 | 10 | 6 | 10 | 0.2113 | 0.0365 | 5.79 | 0.0000 |
| 2 | 10 | 8 | 10 | 0.2246 | 0.0365 | 6.16 | 0.0000 |
| 2 | 1 | 2 | 3 | -0.1028 | 0.0365 | -2.82 | 0.0336 |
| 2 | 1 | 2 | 5 | -0.1991 | 0.0365 | -5.46 | 0.0000 |
| 2 | 1 | 2 | 10 | -0.4409 | 0.0365 | -12.09 | 0.0000 |
| 2 | 3 | 2 | 10 | -0.3382 | 0.0365 | -9.27 | 0.0000 |
| 2 | 5 | 2 | 10 | -0.2419 | 0.0365 | -6.63 | 0.0000 |
| 4 | 1 | 4 | 5 | -0.1691 | 0.0365 | -4.64 | 0.0000 |
| 4 | 1 | 4 | 10 | -0.3154 | 0.0365 | -8.65 | 0.0000 |
| 4 | 3 | 4 | 5 | -0.1042 | 0.0365 | -2.86 | 0.0300 |
| 4 | 3 | 4 | 10 | -0.2504 | 0.0365 | -6.87 | 0.0000 |
| 4 | 5 | 4 | 10 | -0.1463 | 0.0365 | -4.01 | 0.0006 |
| 6 | 1 | 6 | 10 | -0.2725 | 0.0365 | -7.47 | 0.0000 |
| 6 | 3 | 6 | 10 | -0.2233 | 0.0365 | -6.12 | 0.0000 |
| 6 | 5 | 6 | 10 | -0.1850 | 0.0365 | -5.07 | 0.0000 |
| 8 | 1 | 8 | 3 | -0.1166 | 0.0365 | -3.20 | 0.0108 |
| 8 | 1 | 8 | 5 | -0.1654 | 0.0365 | -4.53 | 0.0000 |
| 8 | 1 | 8 | 10 | -0.3163 | 0.0365 | -8.67 | 0.0000 |
| 8 | 3 | 8 | 10 | -0.1997 | 0.0365 | -5.47 | 0.0000 |
| 8 | 5 | 8 | 10 | -0.1509 | 0.0365 | -4.14 | 0.0006 |

Data points displayed are the statistically significant interactions of the 48 (of 120) important comparisons between volumes and layers. Heavy gridline separates retaining volume while comparing layers (above) from retaining layers while comparing volume (below).

[a]Values of P < 0.05 are considered significant.

LSM = least square means

SE = standard error

t = T-Statistic to Test H0: Diff = 0

laparotomy sponges was successful in decreasing the resulting area of bacterial colonization, but not eliminating it. Interestingly, increasing the number of layers from six to eight did not provide any further protection from contamination.

There was no statistically significant relationship when looking at all 120 comparisons for the interaction of layer-density and volume. However, 25 of these comparisons were statistically significant. This data was further evaluated, and a predictable relationship was established. As the number of layers of sponges increases, the infected area decreases in a similar linear fashion across all volumes. No conclusions can be drawn from comparison of this interaction due to its lack of overall significance. The interactions between sponge layer-density and volume of contaminant should be further investigated.

Growth occurring on the plates for all investigated factors was an unexpected finding. This suggests that laparotomy sponges may not offer as much protection from low-viscosity fluids that carry microbial populations as was previously assumed. As such, the results of the present study show that the use of laparotomy sponges may offer significant, though incomplete, protection from overflow GI fluid of any volume and should help guide the surgeon in appropriate follow-up decontamination procedures (i.e., peritoneal lavage, local visceral lavage, etc.). Although the two studies evaluating peritoneal lavage in veterinary patients are based on patients with septic peritonitis, we can extrapolate that with a large volume of overflow GI contents in non-septic patients, peritoneal lavage is indicated to decrease bacterial load and/or resistance status [13, 14]. Unfortunately, the amount of sterile lavage fluid remains in question and is outside the scope of this study.

Outcomes of GI surgical procedures are dependent on a multitude of patient and surgical factors. Factors that can be controlled by the surgical team include the use of aseptic technique and decision making to maintain sterility. Specifically, surgeon preference drives perioperative antibiotic selection, how laparotomy sponges are applied to the incision, the method of isolating the viscera, and the numbers of sponges used for these purposes. Patient factors, however, cannot be controlled by the surgical team. Patient size can limit the space available for sponges to be placed as desired within the abdominal cavity. Contents within the GI viscera can overflow upon incision and/or closure of the hollow viscus, despite all efforts to remove contents from the surgical site using methods such as orogastric tube passage, massage orad or aborad, use of atraumatic intestinal forceps, and the most steady-handed assistant. In the living patient, additional factors to consider are comorbidities that may impact the immune system and its responses. It is therefore the responsibility of the surgeon at hand to minimize the contamination to the best of their abilities. While changing gloves and instruments are part of standard protocols prior to proceeding with other procedures or abdominal wall closure, there are no published guidelines to laparotomy sponge use or decontamination following visceral closure.

There are limitations to this study. The in vitro nature of this study prevents investigation of the immune system that assists in vivo. In a healthy patient without co-morbidities, leakage of 1ml of GI contents onto six or eight layers of laparotomy sponges may not need any further decontamination although in a patient with co-morbidities and these same variables, peritoneal lavage may be indicated. Furthermore, perioperative antibiotics and their effects to reduce or eliminate bacterial inoculation could not be evaluated. Investigation is warranted in this area. The GI microflora contains an abundant variety of bacterial species of varying pathogenicity and virulence, but bacterial inoculation in our study was limited to a single non-pathogenic strain of bacteria. This study was performed using inoculum water of a single turbidity and viscosity while GI contents consist of variable viscosity, digestible particulate matter, and may contain foreign materials. This could alter the magnitude of wicking through the fibers of laparotomy sponges and therefore alter the motility of microbes through sponges. Similarly, this study was performed using laparotomy sponges of the same fiber type, weave, and cross-

stitching, thus these results should only be anticipated for products of similar manufacturing. Data acquisition was performed using macroscopically visible bacterial colonies on the surface of the blood agar medium, producing results of two-dimensional data rather than three-dimensional data. The additional dimension of depth of bacterial colony penetration into the medium was not possible due to the physical limitations of the plate exterior and visibility of contents in this dimension.

In conclusion, increasing the layer-density of laparotomy sponges and decreasing the volume of *E. coli* water were both significant in decreasing strikethrough bacterial contamination in vitro. However, increasing the density from six layers to eight layers does not offer a statistically significant additional protection to underlying structures. While further research is needed for in vivo guidelines, this study can be used by any surgeon performing GI surgery to guide their laparotomy sponge use. The authors recommend a minimum of 6 layers of laparotomy sponges when performing GI surgery to help reduce bacterial contamination.

## Supporting information

**S1 Fig. Experimental setup.**
(TIF)

**S2 Fig. Examples of blood agar plates photographed after 48 hours of incubation after inoculation with *E. coli* water.** A–Trial 1 for 2 Layers. B–Trial 1 for 4 Layers. C–Trial 1 for 6 Layers. D–Trial 1 for 8 Layers.
(TIF)

**S3 Fig. Screenshots of procedure for surface area measurements.** A–infected area in pixels, and B–entire blood agar medium on plate in pixels.
(TIF)

## Acknowledgments

Preparation of experimental material (*E. coli* water*)* was performed at the Veterinary Laboratory of Rochester. The authors thank Paula Hilling and Victoria Small for their assistance with *E coli* water concept design and assistance with equipment.

## Author Contributions

**Conceptualization:** Alla Bezhentseva, Daniel E. Hoffmann.

**Investigation:** Alla Bezhentseva.

**Project administration:** Lindsay L. St. Germaine, Daniel E. Hoffmann.

**Supervision:** Lindsay L. St. Germaine, Daniel E. Hoffmann.

**Writing – original draft:** Alla Bezhentseva.

**Writing – review & editing:** Alla Bezhentseva, Lindsay L. St. Germaine, Daniel E. Hoffmann.

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
