## [Decision Letter · Decision Letter 0]

29 Mar 2022

PONE-D-22-07606Efficacy of laparotomy sponges to reduce bacterial contamination using an in vitro gastrointestinal surgery model.PLOS ONE

Dear Dr. Bezhentseva,

Thank you for submitting your manuscript to PLOS ONE. After careful consideration, we feel that it has merit but does not fully meet PLOS ONE’s publication criteria as it currently stands. Therefore, we invite you to submit a revised version of the manuscript that addresses the points raised during the review process. The reviewers expressed great enthusiasm for the work and suggested only very minor clerical changes prior to acceptance.

We look forward to receiving your revised manuscript.

Kind regards,

Christopher Staley, Ph.D.

Academic Editor

PLOS ONE

Journal Requirements:

Reviewers' comments:

Reviewer's Responses to Questions

**Comments to the Author**

1. Is the manuscript technically sound, and do the data support the conclusions?

Reviewer #1: Yes

Reviewer #2: Yes

2. Has the statistical analysis been performed appropriately and rigorously? 

Reviewer #1: Yes

Reviewer #2: Yes

3. Have the authors made all data underlying the findings in their manuscript fully available?

Reviewer #1: Yes

Reviewer #2: Yes

4. Is the manuscript presented in an intelligible fashion and written in standard English?

Reviewer #1: Yes

Reviewer #2: Yes

5. Review Comments to the Author

Reviewer #1: This is the first in vitro study assessing the efficacy of laparotomy sponges of various layer-densities tested at different contamination volumes and during different exposure times in the reduction of E. coli contamination.

The study provides novel information for veterinarians or surgeons who perform gastrointestinal surgery, to which authors recommed to use laparotomy songes with a minimum of 6 layers.

My only suggestion before publication is to reduce the introduction and to move figure legends and tables at the end of the manuscript.

Reviewer #2: Minimizing the risk of bacterial contamination in laparotomies is extremely important, due to the risk of peritonitis and patient death. The primary purpose of the present study was to evaluate the efficacy of laparotomy sponges in reducing bacterial contamination in an in vitro model and to evaluate whether bacterial contamination is dependent upon layer-density of laparotomy sponges or volume of contaminant.

But some points need to be clarified.

1. Methods:

- Experimental procedure – page 13 – lines 144-148: “…sponges were soaked, squeezed out manually, and each folded into the layer-density being evaluated per trial. For 2 layers, 1 sponge was folded in half once. For 4 layers, 1 sponge was folded in half twice. For 6 layers, 1 sponge was folded in half twice and a second sponge was folded in half once and these were stacked. For 8 layers, 2 sponges were each folded in half twice and these were stacked.” It was not clear whether only one side of the sponges or both sides were seeded onto the agar Petry plates.

2. Results:

- Page 17 - Table 1 - put the meaning of SE.

6. PLOS authors have the option to publish the peer review history of their article (what does this mean?). If published, this will include your full peer review and any attached files.

Reviewer #1: No

Reviewer #2: No

---

## [Author Response · Author response to Decision Letter 0]

3 Apr 2022

Editor comments:

The requested changes to the manuscript type have been completed following the style templates provided.

The authors have reviewed all references and cited articles and none were found to be retracted.

As requested, all figure files were uploaded to PACE for evaluation and PACE approved figure files were uploaded with the revised manuscript documents.

Reviewer #1 comments:

Thank you, Reviewer #1. The introduction has been reduced in length. The authors would like to comply with this reviewer’s request to move figure legends and tables to the end of the manuscript. However, the PLOS ONE manuscript style requirements for submission require that figure captions are inserted “immediately following the paragraph in which the figure is first cited,” and the same applies for all tables.

Reviewer #2 comments:

Thank you for your comment and bringing this to the authors’ attention. I have clarified these points in the revised manuscript lines 205-209. Only one surface of the folded and/or stacked laparotomy sponges is in contact with the blood agar to model contact how a laparotomy sponge would contact abdominal viscera during surgical procedures. 

The meaning of SE has been added to the table legend.

---

## [Editor Report · Decision Letter 1]

6 Apr 2022

Efficacy of laparotomy sponges to reduce bacterial contamination using an in vitro gastrointestinal surgery model.

PONE-D-22-07606R1

Dear Dr. Bezhentseva,

We’re pleased to inform you that your manuscript has been judged scientifically suitable for publication and will be formally accepted for publication once it meets all outstanding technical requirements.

Kind regards,

Christopher Staley, Ph.D.

Academic Editor

PLOS ONE
---

## [Editor Report · Acceptance letter]

21 Apr 2022

PONE-D-22-07606R1 

Efficacy of laparotomy sponges to reduce bacterial contamination using an in vitro gastrointestinal surgery model 

Dear Dr. Bezhentseva:

I'm pleased to inform you that your manuscript has been deemed suitable for publication in PLOS ONE. Congratulations! Your manuscript is now with our production department. 

Kind regards, 

on behalf of

Dr. Christopher Staley 

Academic Editor

PLOS ONE